# Mapping the determinants of catalysis and substrate specificity of the antibiotic resistance enzyme CTX-M β-lactamase

Allison Judge [1], Liya Hu[1], Banumathi Sankaran [2], Justin Van Riper[3], B. V. Venkataram Prasad [1] & Timothy Palzkill [1,4✉]

CTX-M β-lactamases are prevalent antibiotic resistance enzymes and are notable for their ability to rapidly hydrolyze the extended-spectrum cephalosporin, cefotaxime. We hypothesized that the active site sequence requirements of CTX-M-mediated hydrolysis differ between classes of β-lactam antibiotics. Accordingly, we use codon randomization, antibiotic selection, and deep sequencing to determine the CTX-M active-site residues required for hydrolysis of cefotaxime and the penicillin, ampicillin. The study reveals positions required for hydrolysis of all β-lactams, as well as residues controlling substrate specificity. Further, CTX-M enzymes poorly hydrolyze the extended-spectrum cephalosporin, ceftazidime. We further show that the sequence requirements for ceftazidime hydrolysis follow those of cefotaxime, with the exception that key active-site omega loop residues are not required, and may be detrimental, for ceftazidime hydrolysis. These results provide insights into cephalosporin hydrolysis and demonstrate that changes to the active-site omega loop are likely required for the evolution of CTX-M-mediated ceftazidime resistance.

[1] Department of Biochemistry and Molecular Biology, Baylor College of Medicine, Houston, TX, USA. [2] Department of Molecular Biophysics and Integrated Bioimaging, Berkeley Center for Structural Biology, Lawrence Berkeley National Laboratory, Berkeley, CA, USA. [3] Graduate Program in Chemical, Physical, and Structural Biology, Baylor College of Medicine, Houston, TX, USA. [4] Department of Pharmacology and Chemical Biology, Baylor College of Medicine, Houston, TX, USA. ✉email: timothyp@bcm.edu

β-lactamases pose a grave threat to the efficacy of β-lactam drugs, which are the most commonly prescribed class of antibiotics[1]. β-lactamases, which are often encoded by mobile plasmids, are able to hydrolyze the amide bond of the β-lactam ring, rendering these antibiotics ineffective. β-lactamases fall into four classes (A, B, C, and D), based on amino acid sequence homology and function[2,3]. Bacterial resistance to penicillin and early-generation cephalosporin antibiotics due to β-lactamases led to the development of oxymino-cephalosporins, such as cefotaxime and ceftazidime, which are less susceptible to hydrolysis. The widespread use of these antibiotics, however, has led to the emergence of extended-spectrum β-lactamases (ESBLs), which are defined by their ability to hydrolyze oxymino-cephalosporins[4].

The CTX-M family of β-lactamases are the most commonly found ESBLs in Gram-negative bacteria and are notable for their ability to rapidly hydrolyze cefotaxime, but not ceftazidime[4,5]. While ceftazidime is generally a poor substrate for CTX-M enzymes, variants of CTX-M have evolved with increased activity against the antibiotic[6–9]. The CTX-M family are divided into five clusters based on sequence homology: CTX-M-1, CTX-M-2, CTX-M-8, CTX-M-9, and CTX-M-25, with each cluster named based on the archetypal member of each group[9]. The enzyme used in this study, CTX-M-14, is a member of the CTX-M-9 subgroup. CTX-M-14 has served as a model system for studies of CTX-M enzyme structure and mechanism[10–12]. CTX-M enzymes, like other class A enzymes, utilize a catalytic serine residue (Ser70) to attack the carbonyl carbon of the β-lactam ring, leading to the formation of a covalent, acyl-enzyme intermediate[13,14]. In a subsequent deacylation step, residue Glu166 serves as a general base, activating a water molecule that attacks the carbonyl carbon of the ester bond of the acyl enzyme, releasing the hydrolyzed substrate and regenerating the enzyme[15–17].

Beyond the catalytic Ser70 and Glu166 residues, other active site residues contribute to enzyme activity through substrate binding, facilitating proton transfer, or orienting catalytic residues. An omega loop structure that includes Glu166, Pro167, and Asn170, forms the floor of the active site (Fig. S1a). Pro167, present in a *cis*-peptide bond configuration, orients the loop to position Glu166 for efficient deacylation of penicillins and cefotaxime, but is found to be mutated in some ceftazidime-resistant CTX-M clinical isolates[7,18–20]. Asn170 acts along with Glu166 to coordinate a water molecule within the active site to facilitate the acylation and deacylation steps[21–24]. The VNYN (103–106) loop contains residues Asn104,

Tyr105, and Asn106 and is adjacent to the substrate binding pocket (Fig. S1b). In cefotaxime hydrolysis, Asn104 has been shown to form a hydrogen bond with the substrate and facilitate cefotaxime hydrolysis, and Tyr105 interacts with cefotaxime through hydrophobic interactions[25]. Meanwhile, Asn106 participates in a hydrogen bond network, including an interaction with the main chain of Val103, to orient the Asn104 side chain into the active site[25]. This VNYN loop is conserved within CTX-M enzymes and is considered critical for ESBL activity[10,23,26]. Residues Lys73, Ser130, and Lys234 participate in a proton shuttle to facilitate acylation[24,27]. Asn132, a highly conserved residue, forms a hydrogen bond with the substrate to facilitate binding and stabilize the transition state[28]. The β3 strand, across the active site from the omega loop, contains residues Lys234, Thr235, Gly236, Ser237, Gly238, and Asp240 (Fig. S1c). Of these, Lys234, Thr235, and Ser237 are thought to facilitate substrate binding[4,29–31]. It is likely that Gly236 and Gly238 are crucial in CTX-M enzymes because a side chain at either position would point toward the Ser70 helix behind the β3 strand, disrupting the strand's conformation and potentially protein folding. Arg276 is conserved across CTX-M enzymes, but is not found in other extended-spectrum β-lactamases[10]. A previous mutagenesis study suggests that Arg276 is not strictly required for catalytic activity in CTX-M enzymes, despite its sequence conservation[32].

Here, we identify the CTX-M residues required for the catalysis of several β-lactam antibiotics, including ampicillin, cefotaxime, and ceftazidime, to further understand the drivers of substrate specificity and resistance. Seventeen residues within the active site (Fig. 1), including those outlined above, were randomized using site-directed mutagenesis. The resulting random libraries were introduced into *E. coli* and selected for function against ampicillin and cefotaxime—both of which are efficiently hydrolyzed by CTX-M-14—as well as ceftazidime, which is poorly hydrolyzed by CTX-M enzymes. Following antibiotic selection, the surviving clones were pooled, and deep sequencing was used to determine the relative frequency of each amino acid at each active site position. This revealed active site positions where the wild-type residue is strictly required for high-level function against cefotaxime and ampicillin (Ser70, Lys73, Ser130, Asn132, Asn170, Glu166, and Gly236), suggesting these residues contribute to enzyme function across classes of β-lactams, as well as residues that are substitutable (Tyr105, Asn106, Gly238, Asp240, and Arg276), indicating a less critical contribution to catalysis. Additionally, differential sequence requirements for hydrolysis of ampicillin versus cefotaxime revealed the residues controlling substrate specificity (Pro167, Asn170, Lys234, Thr235, and Ser237). Finally, we determined that the sequence requirements for ceftazidime hydrolysis largely followed those of cefotaxime, with the exception that key omega loop residues (Glu166, Pro167, and Asn170) are not required and may, in fact, be detrimental for ceftazidime hydrolysis. These results provide insights into the unique ability of CTX-M enzymes to hydrolyze extended-spectrum cephalosporins.

## Results and discussion

**Deep sequencing of CTX-M-14 randomized codon libraries.** Based on X-ray crystal structures of CTX-M enzymes, 17 active site residues, which include all positions that can interact directly with the substrate as well as several second shell residues, were targeted for mutagenesis. Randomized single-codon libraries were constructed for each of the 17 positions. To assess the amino acid sequence requirements for function, each library was introduced into *E. coli*, and the resulting clones were selected for function by growth in media containing either ampicillin, cefotaxime, or ceftazidime. The distribution of amino acid types in the population of functional clones after each selection was assessed by next-generation sequencing (NGS) to determine the frequency of

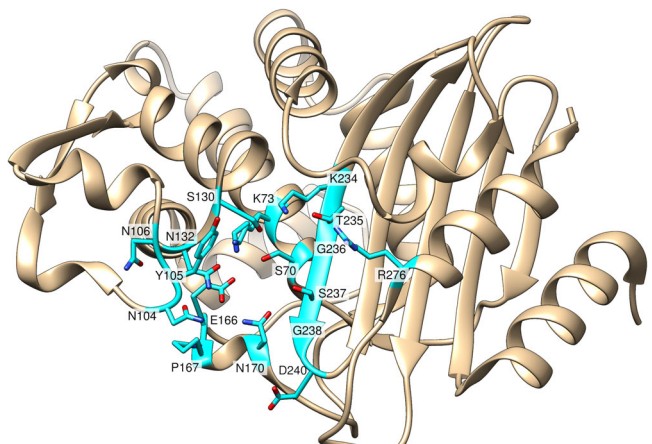

**Fig. 1 X-ray crystal structure (PDB: 1YLT) of CTX-M-14 β-lactamase.** Using the previously published X-ray crystal structure of CTX-M-14 (PDB: 1YLT), 17 active site residues selected for randomization and deep sequencing are highlighted in cyan. Heteroatoms of amino acid side chains are represented in red (oxygen) and blue (nitrogen).

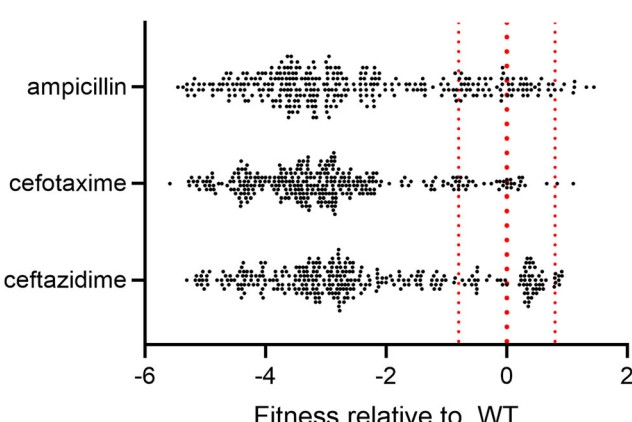

## Single mutant fitness distribution

**Fig. 2 Distribution of single mutant fitness values relative to CTX-M-14$^{WT}$.** The red lines represent the wild-type fitness value ($F = 0$) and the 95% confidence interval above and below. Mutants that increase CTX-M-14 fitness have $F > 0.8$ and those that decrease fitness have $F < 0.8$.

amino acids relative to the wild-type amino acid. For comparison, we also determined the frequency of each amino acid in the naïve library pools before β-lactam antibiotic selection.

The frequency of each possible amino acid substitution was compared to the frequency of the wild-type amino acid and was used to calculate the relative fitness of each amino acid substitution conferred to *E. coli* after antibiotic selection (Eq. 1)[33].

$$F = \log_{10}\left[\frac{N_{mut}^{sel}}{N_{mut}^{naïve}}\right] - \log_{10}\left[\frac{N_{WT}^{sel}}{N_{WT}^{naïve}}\right] \quad (1)$$

The first term of Eq. 1 accounts for the frequency of a given amino acid ($N_{mut}^{sel}$) within an antibiotic-selected library and its frequency in the naïve library ($N_{mut}^{naïve}$). The second term accounts for the frequency of the wild-type amino acid before ($N_{WT}^{naïve}$) and after selection ($N_{WT}^{sel}$). Within each library, the wild-type enzyme has a relative fitness value of $F = 0$, while a deleterious amino acid substitution has a fitness of $F < 0$, and a beneficial substitution has a fitness of $F > 0$. In this way, the analysis provides a quantitative estimate of the relative levels of function provided by each amino acid substitution at a given active site position for a given antibiotic selection. Generally, mutations to CTX-M-14 were found to be detrimental, with 97% (313/323 mutations) leading to a significant decrease in *E. coli* fitness for at least one antibiotic (Fig. 2). Detailed figures can be found in the supplementary information, which depicts the fitness of each mutated enzyme against ampicillin (Fig. S2), cefotaxime (Fig. S3), and ceftazidime (Fig. S4). A comparison of the results between antibiotics reveals positions where the wild-type amino acid is required for catalytic activity towards all β-lactam antibiotics versus positions that exhibit differential sequence requirements between antibiotics, i.e., substrate specificity determinants, and finally, positions where the identity of the amino acid is not required for function, regardless of the antibiotic.

To quantify and visualize the differing sequence stringency between substrates, sequence logos were generated, where the size of the single letter amino acid code represents its frequency following *E. coli* selection under a given antibiotic (Fig. 3). Additionally, the $k^*$ value, or effective number of substitutions, was calculated for each residue following each antibiotic selection. A $k^* = 1$ indicates that only one amino acid residue appears at a given position following antibiotic selection, while a $k^* = 20$ indicates that all amino acids are present in equal frequencies

following selection[34,35]. Figure 4 shows the $k^*$ value of each active site position under each antibiotic selection. We categorized the residue positions based on $k^*$ values as having stringent (red, less than 1.5 effective substitutions), flexible (yellow, 1.5–4.0 effective substitutions), or loose (green, greater than 4.0 effective substitutions) sequence requirements.

**Key catalytic residues are essential for penicillinase and ESBL activity.** Based on our deep sequencing results of the library selections, there is a set of core residues in CTX-M-14 that are essential for activity towards ampicillin and cefotaxime, i.e., any substitution leads to a significantly reduced fitness relative to wildtype, $F < 0$ (Figs. S2, S3) and their effective number of substitutions ($k^* \leq 1.5$) indicate high stringency (Fig. 4). These core residues include Ser70, Lys73, Ser130, Asn132, Glu166, and Gly236. The S130T, E166Y, and N132S mutations are exceptions to the otherwise stringent sequence requirements; based on their relative fitness, S130T and E166Y provide some function against cefotaxime (Fig. S3), while N132S provides some function against ampicillin (Fig. S2). Additionally, the wild-type Glu166 was not enriched after ceftazidime selection, as discussed in the following section on differential ceftazidime sequence requirements. Otherwise, the antibiotic-selected libraries for Ser70, Lys73, Ser130, Asn132, Glu166, and Gly236 returned overwhelmingly wild-type sequences, as visualized in the sequence logo (Fig. 3). These results indicate that *E. coli* containing the wild-type amino acid were far more likely to survive in the presence of β-lactam antibiotic than those containing a mutation at any of these six residue positions.

The S130T mutant, surprisingly, was predicted to retain activity against cefotaxime, but lose penicillinase activity. The mutant enzyme was purified and tested for steady-state kinetics parameters, and the results support the predicted activity of S130T against cefotaxime, in that $k_{cat}$ is similar to wildtype and $k_{cat}/K_M$ is higher than that observed for the wild-type enzyme (Table 1). The E166Y mutant, when purified, however, did not retain wild-type levels of activity toward cefotaxime. The mutation results in a 30,000-fold drop in $k_{cat}$ and a 2000-fold drop in $k_{cat}/K_M$ (Table 1). The retention of some activity against the substrate is notable and likely responsible for the elevation of its fitness above other Glu166 mutations (Fig. S3), as demonstrated previously in the Class A β-lactamases, TEM-1[36,37] and PenP[38].

The N132S mutation was not individually mutated in this study, but fitness results suggest that it has partial function in ampicillin hydrolysis (Fig. S2). Previously, Asn132 mutations were made in the Class A β-lactamase, *Streptomyces albus* G[28]. The mutation of Asn132 to alanine led to a steep decrease in catalytic efficiency ($k_{cat}/K_M$) for ampicillin to 0.3% of wild-type activity, driven primarily by a drop in $k_{cat}$. The catalytic efficiency was partially recovered in the N132S mutant to 16% of wild-type activity. It was concluded that Ser132 retains partial activity due to its ability to form an H-bond with penicillin substrates that provide optimal geometry for nucleophilic attack by Ser70 during the transition state of the acylation step. Cephalosporin hydrolysis, however, requires more precise geometry that can only be provided by the wild-type asparagine residue at position 132. Our results are consistent with conclusions from this study in *Streptomyces albus* G β-lactamase—that Ser132 retains partial function for Class A β-lactamases in penicillin hydrolysis, but Asn132 is preferred for penicillin and critical for cephalosporin hydrolysis.

Aside from the outlined exceptions, our results align with the current understanding of these six amino acids within the active site and confirm that their presence is crucial for resistance to penicillins and the cephalosporin cefotaxime in CTX-M enzymes[6,7,16,24,27,28].

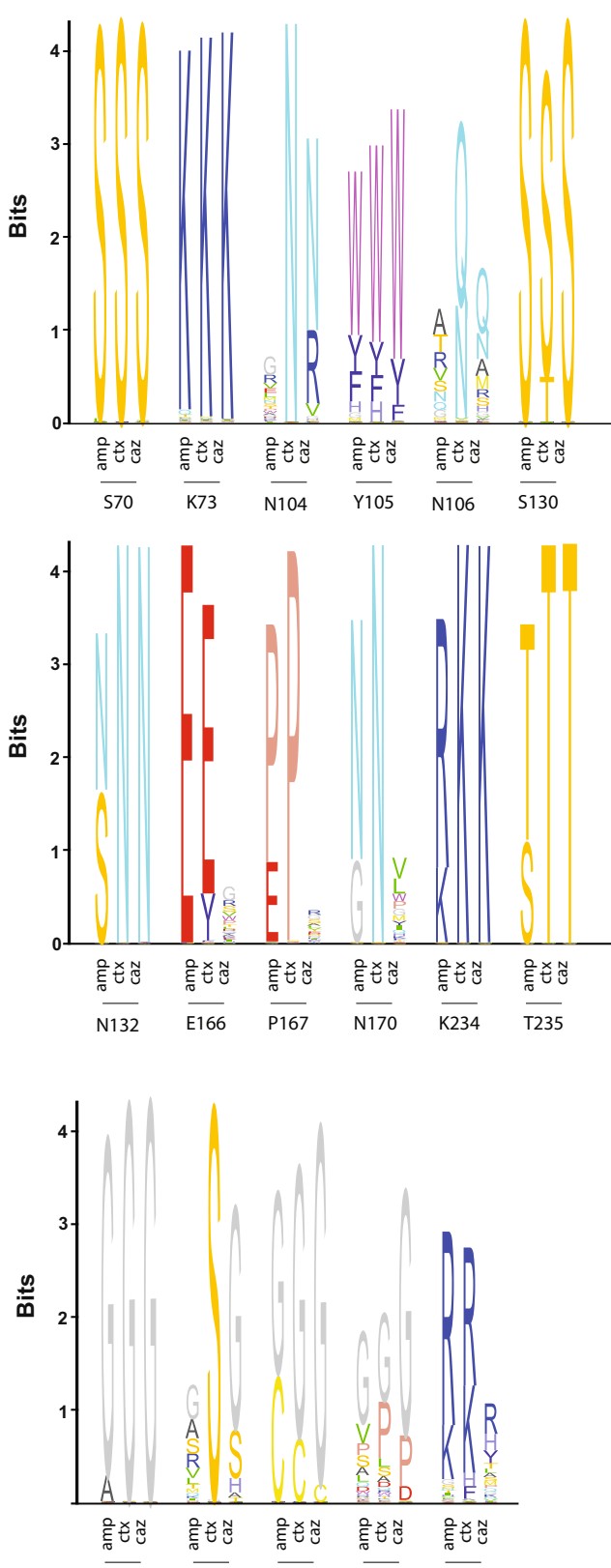

**Fig. 3 Sequence logos representing the relative frequency of each residue at each of the 17 randomized active site positions.** The height of each amino acid (single letter code) is proportional to its frequency following selection in ampicillin (amp), cefotaxime (ctx), or ceftazidime (caz).

**A subset of positions can be substituted and retain penicillin and extended-spectrum cephalosporin resistance activity.** A handful of residues—Tyr105, Asn106, Gly238, Asp240, and Arg276—can tolerate some substitutions and retain high levels of ampicillin and cefotaxime hydrolysis. Amino acid substitutions that are predicted to have high catalytic activity based on the sequencing results were introduced at positions Tyr105, Asn106, and Asp240, and the modified enzymes were purified. Steady-state enzyme kinetic parameters and in vivo minimum inhibitory concentrations (MIC) were determined to further validate the sequencing results displayed in Fig. 3.

Tyr105, which makes hydrophobic interactions with β-lactam substrates once bound[25], can be substituted for other aromatic side chains—tryptophan or phenylalanine—with no predicted impact on enzyme activity and resistance (Fig. 3). The antibiotic resistance levels of E. coli containing the Y105W mutant were tested, and the mutant exhibited no significant change (defined as a fourfold or larger change in MIC) against ampicillin or cefotaxime (Table 2). Moreover, the turnover rate ($k_{cat}$) and catalytic efficiency ($k_{cat}/K_M$) for both substrates were similar to those of the wildtype (Tables 1, 3). Consistent with NMR results published on the Y105W mutation in TEM-1 β-lactamase[39], our X-ray crystal structure of the mutant showed virtually no change in the active site apart from the tryptophan substitution itself (Fig. 5 and Table 4). These results show that there is a strong preference for aromatic amino acids at position 105, consistent with the need for a flat, hydrophobic surface to form a wall of the active site[40]. Presumably, other hydrophobic residues, such as leucine and isoleucine, do not provide function due to steric constraints with a bound substrate.

Deep sequencing of the Asn106 library following ampicillin selection revealed a wide array of functional substitutions ($k* = 8.7$), indicating many different amino acids are consistent with high levels of ampicillin activity (Fig. 4 and S2). In contrast, sequencing of the library after cefotaxime selection indicated that glutamine was the only substitution that appeared as frequently as the wild-type asparagine (Fig. 3 and S3). Asparagine and glutamine are isosteric and differ only in the length of the hydrocarbon side chain. We previously showed that Asn106 forms a hydrogen bond network that maintains the 103–106 loop in a conformation that facilitates cefotaxime hydrolysis[25]. Steady-state kinetics for the N106Q mutant showed that the catalytic efficiency ($k_{cat}/K_M$) of N106Q for cefotaxime hydrolysis is approximately 1.5-fold that of wildtype, consistent with our deep sequencing results (Table 1). Additionally, steady-state kinetics showed that the turnover rate ($k_{cat}$) and catalytic efficiency ($k_{cat}/K_M$) of the mutant enzyme were at wild-type levels for ampicillin hydrolysis (Table 1), as expected since our sequencing results suggest that N106Q is a functional enzyme for ampicillin hydrolysis. The MIC of N106Q was unchanged for both ampicillin and cefotaxime (Table 2). We previously demonstrated that the natural variant, N106S, exhibits reduced cefotaxime hydrolysis, consistent with our finding here that Ser106 is not a functional substitution after cefotaxime selection[25]. In a subsequent study, it was shown that the CTX-M-14 N106S mutation is responsible for enhanced inhibition potency of the β-lactamase inhibitory protein[41], which is due to a change in conformation of the 103–108 loop. The conformational change is also the source of decreased cefotaxime hydrolysis by the N106S enzyme. The N106Q mutant, however, retains catalytic activity for cefotaxime, suggesting that a longer glutamine side chain can be accommodated without changing the 103–106 loop conformation.

Deep sequencing of the Asp240 CTX-M-14 library shows substantial sequence variation, with loose sequence requirements after selection with both ampicillin and cefotaxime (Fig. 4).

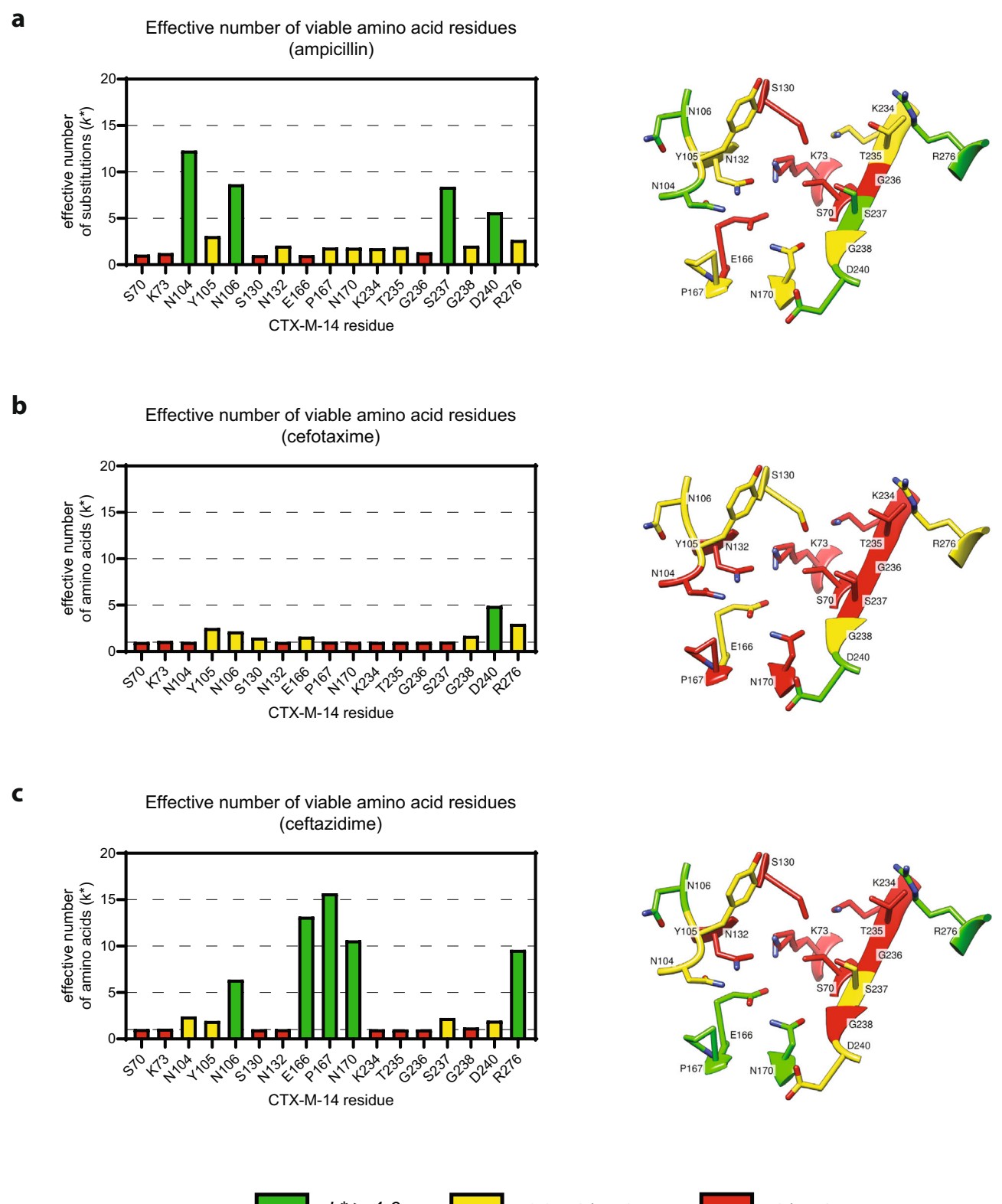

**Fig. 4 The effective number of substitutions for each CTX-M-14 active site position.** The $k^*$ bar plot in each panel represents the effective number of substitutions for each randomized active site position when selected for **a** Ampicillin resistance, **b** Cefotaxime resistance, and **c** Ceftazidime resistance, as calculated according to Eq. 2 (Methods). Each bar is color-coded based on its $k^*$ value so that residue positions that have only one functional amino acid ($k^* < 1.5$) are red, residues that have a conservative number of potential substitutions ($1.5 < k^* < 4.0$) are yellow, and residues that can accommodate a wide variety of substitutions ($k^* > 4.0$) are green. The crystal structure of wild-type CTX-M-14 is shown on the right of each panel (PBD: 1YLT) and is color-coded by the same metric to display where each residue lies within the active site.

**Table 1 Steady-state kinetic parameters of CTX-M-14 hydrolysis of ampicillin and cefotaxime.**

| CTX-M-14 | Ampicillin | | | Cefotaxime | | |
|---|---|---|---|---|---|---|
| | $k_{cat}$, s$^{-1}$ | $K_M$, µM | $k_{cat}/K_M$, µM$^{-1}$ s$^{-1}$ | $k_{cat}$, s$^{-1}$ | $K_M$, µM | $k_{cat}/K_M$, µM$^{-1}$ s$^{-1}$ |
| WT | 55 ± 2[a] | 33 ± 4 | 1.7 ± 0.2[b] | 76 ± 3 | 105 ± 11 | 0.72 ± 0.08 |
| Y105W | 66 ± 6 | 51 ± 10 | 1.3 ± 0.3 | 54 ± 3 | 71 ± 9 | 0.76 ± 0.1 |
| N106Q | 27 ± 1 | 18 ± 2 | 1.5 ± 0.2 | 96 ± 3 | 87 ± 8 | 1.1 ± 0.1 |
| S130T | ND | >2000 | $7.5 \times 10^{-4} \pm 5 \times 10^{-6}$ | 88 ± 10 | 12 ± 4 | 7.8 ± 0.4 |
| E166Y | $1.5 \times 10^{-3} \pm 1.6 \times 10^{-4}$ | 8.4 ± 4 | $1.8 \times 10^{-4} \pm 0.5$ | $2.4 \times 10^{-3} \pm 5 \times 10^{-5}$ | 7.0 ± 0.7 | $3.5 \times 10^{-4} \pm 0.1$ |
| P167E | 72 ± 6 | 45 ± 7 | 1.6 ± 0.3 | 11 ± 0.5 | 39 ± 5 | 0.28 ± 0.04 |
| P167S[c] | 29 ± 3 | <12 | >2.4 | 300 ± 30 | 37 ± 6 | 8.0 ± 2 |
| K234R[c] | 330 ± 30 | 1100 ± 200 | 3.1 ± 0.3 | 0.2 ± 0.01 | 11 ± 4 | 1.8 ± 0.5 |
| T235S | 27 ± 1 | 54 ± 9 | 0.5 ± 0.09 | 170 ± 40 | 2800 ± 800 | 0.061 ± 0.02 |
| D240G[c] | 640 ± 10 | 75 ± 10 | 8.6 ± 1 | 320 ± 50 | 52 ± 9 | 6.2 ± 1 |
| D240P | 4.5 ± 0.4 | < 5 | ND | 14 ± 1 | 15 ± 3 | 0.93 ± 0.2 |
| D240V | 56 ± 3 | 23 ± 5 | 2.4 ± 0.5 | 77 ± 5 | 45 ± 10 | 1.7 ± 0.4 |

[a]The ± symbol represents the standard error for each value, based on the Michaelis–Menten curve fit.
[b]The standard error for $k_{cat}/K_M$ was calculated by propagating the error of $k_{cat}$ and $K_M$, respectively, using Eq. 3 (Methods).
[c]Previously published data[20,27]

**Table 2 Minimum inhibitory concentrations of CTX-M-14^WT and variants against ampicillin, cefotaxime, and ceftazidime.**

| CTX-M-14 | Ampicillin (ug/mL) | Cefotaxime (ug/mL) | Ceftazidime (ug/mL) |
|---|---|---|---|
| WT | 400 | 32 | 1 |
| Y105W | 400 | 16 | 2 |
| N106Q | 400 | 16 | 1 |
| E166A | ≤6.25 | ≤0.5 | 2 |
| P167E | 200 | 0.5 | 2 |
| T235S | 100 | 0.5 | 0.5 |
| D240P | 200 | 16 | 2 |
| D240V | 200 | 4 | 0.5 |
| pTP123 (no insert) | 1.56 | ≤0.125 | 0.25 |

**Table 3 Steady-state kinetic parameters of CTX-M-14^WT and variants versus ceftazidime.**

| CTX-M-14 | Ceftazidime | | |
|---|---|---|---|
| | $k_{cat}$, s$^{-1}$ | $K_M$, µM | $k_{cat}/K_M$, µM$^{-1}$ s$^{-1}$ |
| WT | ND | >1000 | $5.2 \times 10^{-4} \pm 2 \times 10^{-5}$ |
| Y105W | ND | >1000 | $5.7 \times 10^{-4} \pm 5 \times 10^{-5}$ |
| S130T | ND | >800 | $8.4 \times 10^{-4} \pm 4 \times 10^{-5}$ |
| P167S[c] | 230 ± 4.5 | 15 ± 1.8 | 15 ± 1.8 |
| D240G[c] | ND | >500 | $0.013 \pm 7 \times 10^{-4}$ |

[a]The ± symbol represents the standard error for each value, based on the Michaelis–Menten curve fit.
[b]The standard error for $k_{cat}/K_M$ was calculated by propagating the error of $k_{cat}$ and $K_M$, respectively, using Eq. 3 (Methods).
[c]Previously published data[20].

A previously studied CTX-M variant, D240G, was found to be the most enriched residue at the position following selection in both ampicillin and cefotaxime[20]. Among the remaining substitutions tested here, most of which produced a functional enzyme based on sequencing results (Figs. S2, S3), D240P and D240V were predicted to have the highest activity and were chosen for further validation. Aside from glycine, proline was the most enriched residue after cefotaxime selection, while valine was the most enriched following the ampicillin selection of the D240 library. Steady-state kinetics confirmed the prediction that D240P exhibits a modest increase in catalytic efficiency against cefotaxime but a decrease against ampicillin (Table 1). The MIC results for D240P demonstrate wild-type levels of ampicillin and cefotaxime resistance (Table 2). Meanwhile, steady-state kinetic parameters for the D240V enzyme indicate a twofold increase in catalytic efficiency in cefotaxime hydrolysis and a 1.5-fold increase in catalytic efficiency for ampicillin hydrolysis (Table 1). The MIC of the D240V mutant suggests retention of ampicillin resistance (a modest twofold decrease) and a significant decrease in cefotaxime resistance (an eightfold decrease), consistent with the sequencing results.

Arg276 is conserved across CTX-M enzymes, though not among other β-lactamases[10]. Sequencing results for position 276 indicate that the wild-type arginine, as well as lysine and histidine substitutions, have higher fitness values against ampicillin and cefotaxime than other substitutions (Fig. 3 and S2, S3). A previous mutagenesis study found that the R276H mutation in CTX-M-1 did not lead to significant decreases in MIC for amoxicillin (a penicillin antibiotic) or cefotaxime[32]. However, several other, less conservative mutations at position 276 were detrimental. The authors concluded that, while not pivotal for catalysis, Arg276 is critical for general stability in the CTX-M-1 enzyme. Conservative substitutions to lysine and histidine, may maintain stabilizing hydrogen bond network interactions, explaining the preference for these mutations over others at position 276 demonstrated in CTX-M-1 and here in CTX-M-14.

Gly238 lies within the β3 strand, on the right side of the active site in Fig. 1. The side chains introduced at position 238 would point toward the α helix containing Ser70 (pictured behind Gly238 in Fig. S1c). This would likely disrupt catalytic machinery (located both on the β3 strand and the α helix behind it) and destabilize the enzyme. However, sequencing results suggest partial function for one mutation, G238C (Fig. 3 and S2–S4). The wild-type CTX-M-14 structure (PDB: 1YLT) shows that the Cys69 side chain faces Gly238, and the introduction of the Cys238 mutation could result in a disulfide bond between the two cysteines. For this reason, G238C may not cause the same steric disruption that would be introduced by another side chain at position 238. Further experimentation would be required to confirm that Cys238 forms a disulfide bond with Cys69, but since other small, hydrophobic residues do not have partial function (Fig. 3 and S2–S4), it is likely that the unique ability of cysteine to form a covalent disulfide bond is responsible for the partial fitness of this CTX-M mutant.

**Differential sequence requirements reveal residues essential only for cefotaxime hydrolysis activity.** Seven residue positions were found to be key to cefotaxime resistance but not required

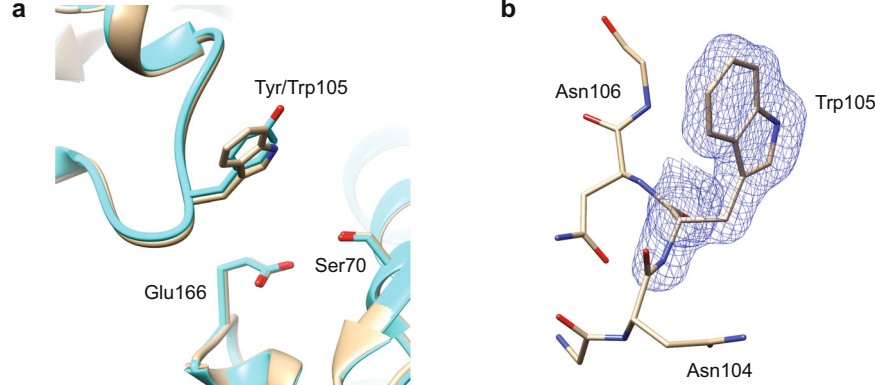

**Fig. 5 CTX-M-14 Y105W enzyme structure. a** CTX-M-14 Y105W enzyme structure (blue, PDB: 7UON) determined at 1.35 Å superimposed on wild-type CTX-M-14 (tan, PDB: 1YLT). Note that the tryptophan assumes an essentially identical position as the wild-type tyrosine. **b** Polder map of 7UON (3.0σ), constructed with Trp105 omitted, which demonstrates the certainty of the positioning of Trp105.

**Table 4 X-ray crystallography data collection and refinement statistics (molecular replacement).**

|  | CTX-M-14$^{Y105W}$ (7UON) |
|---|---|
| **Data Collection** | |
| Space group | P 3$_2$ 2 1 |
| Cell dimensions | |
| $a, b, c$ (Å) | 41.43, 41.43, 230.64 |
| $\alpha, \beta, \gamma$ (°) | 90.00, 90.00, 120.00 |
| Resolution (Å) | 28.32–1.35 (1.42–1.35) |
| $R_{merge}$ | 0.069 (0.553) |
| $I / \sigma I$ | 17.0 (3.4) |
| Completeness (%) | 99.1 (98.1) |
| Redundancy | 12.3 (10.8) |
| **Refinement** | |
| Resolution (Å) | 1.35 |
| No. reflections | 51409 |
| $R_{work}/R_{free}$ | 20.7/24.6 |
| No. atoms | |
| Protein | 1991 |
| Ligand/ion | 5 |
| Water | 252 |
| $B$-factors | |
| Protein | 28.1 |
| Ligand/ion | 23.5 |
| Water | 41.7 |
| R.m.s. deviations | |
| Bond lengths (Å) | 0.0058 |
| Bond angles (°) | 0.86 |

Statistics belong to one crystal.
Values in parentheses are for the highest-resolution shell.

for ampicillin resistance based on fitness values derived from sequencing results. Substitutions were identified at residues Asn104, Pro167, Asn170, Lys234, Thr235, and Ser237 that retain high levels of ampicillin resistance, but drastically decrease cefotaxime resistance. In Fig. 4, the sequence requirements of each of these positions go from loose or flexible under ampicillin selection (panel a) to stringent under cefotaxime selection (panel b). Of the mutants predicted by deep sequencing results to retain activity towards ampicillin while losing activity towards cefotaxime (Figs. S2, S3), P167E and T235S were selected for further validation (Tables 1, 3).

Previous studies suggest that a proline to glutamate substitution at position 167 would inactivate the enzyme due to the loss of the *cis* configuration of the peptide bond between Pro167 and the catalytic residue Glu166[42]. However, our sequencing results indicate that the P167E mutation retains high ampicillin resistance levels (Fig. 3). Steady-state kinetic parameters of P167E for ampicillin support this result, with a similar turnover rate and catalytic efficiency as the wild-type enzyme (Table 1). For cefotaxime, however, the P167E mutant enzyme exhibits a sevenfold decrease in the turnover rate and a threefold decrease in catalytic efficiency compared to wildtype. The MIC of *E. coli* containing P167E against ampicillin decreases only twofold, suggesting retention of activity, but decreases dramatically (64-fold) against cefotaxime (Table 2). These results suggest the P167E substitution alters the structure of the omega loop in a way that is consistent with ampicillin hydrolysis, but not cefotaxime hydrolysis.

The other substrate-specificity mutant validated biochemically, T235S, is an isosteric substitution of threonine to serine. Ser235 is found in many non-ESBL β-lactamases that are incapable of hydrolyzing cefotaxime[17,31]. The finding that Ser235 has a lower fitness after cefotaxime selection suggests that Thr235 is required for high-level cefotaxime hydrolysis. Steady-state kinetic parameters for T235S show a threefold decrease in the catalytic efficiency of the enzyme with ampicillin as substrate compared to the wildtype (Table 1). With cefotaxime as substrate, as predicted by sequencing results, the enzyme exhibits a dramatic decrease (tenfold) in $k_{cat}/K_M$ (Table 1). In the β-lactamase mechanism, $k_{cat}/K_M$ reflects the $K_D$ for substrate binding and the acylation rate; therefore, the serine substitution results in lower binding affinity for cefotaxime and/or a slower acylation rate[43]. The MIC values of *E. coli* with the T235S mutant (Table 2) similarly suggest a modest decrease in ampicillin resistance (fourfold) and a dramatic decrease in cefotaxime resistance (64-fold). These results support the role of Thr235 in cefotaxime binding affinity and/or acylation rate for CTX-M enzymes and indicate that threonine, rather than Ser235, which is commonly observed in other class A enzymes, is required for high levels of cefotaxime hydrolysis.

**Library selections with ceftazidime reveal no sequence preference for omega loop residues Glu166, Pro167, and Asn170.** CTX-M enzymes, although highly efficient for cefotaxime hydrolysis, slowly hydrolyze the extended-spectrum cephalosporin ceftazidime. However, variants of CTX-M enzymes, including P167S and D240G, that more rapidly hydrolyze ceftazidime have evolved in drug-resistant bacteria[7,20]. Our deep sequencing results of ceftazidime-selected pools show that the sequence requirements for ceftazidime hydrolysis are broadly similar to those for cefotaxime, suggesting that these residues are responsible for ESBL activity in CTX-M enzymes (Figs. 3, 4). Notably, however, key active site residues within the omega loop

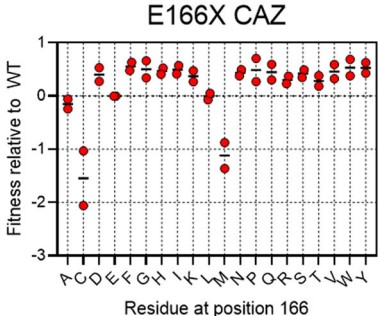 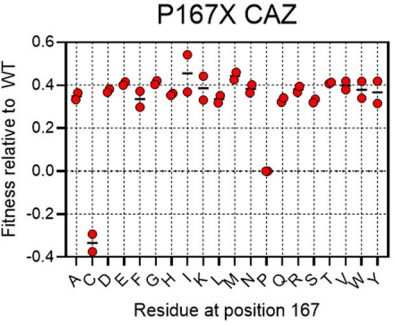 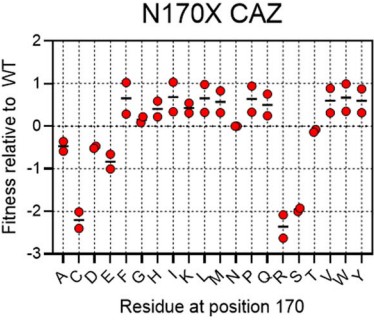

**Fig. 6 Fitness values for omega loop substitutions following ceftazidime selection.** The fitness values of the E166X, P167X, and N170X libraries, based on deep sequencing of library pools after selection for growth in the presence of ceftazidime ($n = 2$). Results show mutants with increased fitness relative to the wildtype after the growth of *E. coli* in the presence of ceftazidime.

(Glu166, Pro167, and Asn170) can be freely substituted and yet retain activity towards ceftazidime, suggesting they do not contribute to hydrolysis (Fig. 6). It has been established that ceftazidime is poorly hydrolyzed by CTX-M and other class A β-lactamases due to steric constraints resulting from the bulky oxyimino side chain that extends into the base of the active site[8]. The oxyimino group clashes with the omega loop that is held rigidly in place by a *cis* peptide bond between Pro167 and Glu166. The *cis* peptide bond is critical in that it places the general base Glu166 in position to extract a proton from water for the deacylation reaction required for hydrolysis of β-lactam antibiotics. Interestingly, nearly all substitutions at Pro167, which mediates the *cis* peptide bond and thus the conformation of the omega loop, results in increased ceftazidime fitness, suggesting that disruption of the omega loop conformation is important for ceftazidime binding and hydrolysis (Fig. 6). Previously, the natural CTX-M-14 variant P167S was shown to increase the flexibility of the loop by eliminating the *cis*-peptide bond between Pro167 and Glu166, resulting in increased flexibility of the active site to accommodate ceftazidime[31,44]. The opening of the loop and consequent movement of Glu166 presumably does not impact deacylation because Glu166 is not contributing to ceftazidime hydrolysis, as evidenced by our sequencing (Fig. 6) and MIC (Table 2) results that show the resistance levels of E166A against ceftazidime are the same as the wildtype. This is likely due to the fact that the deacylation of ceftazidime by Glu166 is highly inefficient[18], leading to no preference for the standard CTX-M mechanism over an alternative, which could use bulk water within the active site to slowly deacylate ceftazidime. It is clear from these results and previous research that CTX-M resistance to ceftazidime could evolve through changes in the omega loop. A limit of this study, however, is that we did not know how omega loop mutations, such as P167S, interact with additional active site mutations. It is known, for example, that the natural mutations P167S and D240G, which each increase ceftazidime hydrolysis individually (Table 3), are antagonistic when combined such that the double mutant has less activity than either single mutant[6,19]. This antagonism may be the source of a more limited range of Pro167 mutations seen in clinical isolates[9].

While substitutions to the omega loop are a requirement for high levels of ceftazidime resistance, changes to Glu166, Pro167, or Asn170 also greatly reduce the activity of CTX-M-14 towards ampicillin and/or cefotaxime. Therefore, evolved enzymes that hydrolyze ceftazidime would tradeoff the ability to hydrolyze penicillins and cephalosporins, suggesting these β-lactams would be effective in the treatment of infections caused by bacteria containing ceftazidime-resistant variants and preventing truly broad-spectrum resistance amongst CTX-M enzymes. Ceftazidime-resistant variants of KPC, a class A carbapenemase, have overcome this tradeoff to

some extent—preserving their original substrate profile in addition to gaining the ability to hydrolyze ceftazidime[45]. This tradeoff is further exacerbated, however, when ceftazidime is combined with the covalent inhibitor of β-lactamase, avibactam. Acquisition of resistance to ceftazidime-avibactam by KPC enzymes generally leads to decreased protein stability and decreased carbapenemase activity in KPC enzymes[46–48]. Rather, ceftazidime-avibactam-resistant β-lactamases must become specialists, abandoning their natural function in a tradeoff for resistance to this antibiotic-inhibitor combination[49]. CTX-M enzymes may experience this same tradeoff.

Aside from the omega loop, another residue where ceftazidime and cefotaxime sequence requirements differ is Ser237. The wild-type serine is clearly preferred over any substitution at position 237 following cefotaxime selection, based on sequencing results (Fig. 3). Under ceftazidime selection, however, His237 and Gly237 appeared more frequently than the wildtype, while Thr237 and Ala237 showed only modest decreases in fitness compared to wildtype (Fig. S3). The wild-type Ser237 is likely preferred for cefotaxime hydrolysis because the serine hydroxyl forms a hydrogen bond with the C4 carboxylate of the substrate[23]. The published structure of ceftazidime bound to the CTX-M-14 E166A/D240G mutant (PDB: 6V7T) demonstrates that the C4 carboxylate of ceftazidime is oriented differently than that of cefotaxime, preventing it from interacting with Ser237. The previously studied P167S mutation expands the active site loop and orients ceftazidime to restore the hydrogen bond between Ser237 and the C4 carboxylate (PDB: 6V8V). This indicates that the alternative orientation of the C4 carboxylate of ceftazidime may change the sequence requirements of Ser237, suggesting that Ser237 could play a more important role if the active site is first expanded through changes to the omega loop, which appears to be an evolutionary path to ceftazidime resistance[31].

The selection of the D240X library against ceftazidime, unsurprisingly, returned D240G as the most viable CTX-M-14 variant (Fig. S3). This naturally-occurring variant has demonstrated increased ceftazidime resistance in other studies, which can act synergistically with other active site mutations to further improve hydrolysis and resistance[7,25,50]. The number of effective substitutions for Asp240 following ceftazidime selection ($k* = 1.9$) further demonstrates this preference for D240G for ceftazidime resistance, in contrast to the higher values for ampicillin ($k* = 5.6$) and cefotaxime ($k* = 4.9$) selection, where position 240 can be substituted for a wider variety of effective residues.

## Conclusions

Through antibiotic selection and deep sequencing of CTX-M-14 mutants, which encompass all possible mutations at 17 active site positions, we have identified residues with stringent sequence

requirements that are required for β-lactamase activity versus all substrates tested (Ser70, Lys73, Ser130, Asn132, and Gly236). We have also identified a residue that can be freely substituted (Asp240) and residues that can adopt a conservative range of mutations (Tyr105, Asn106, Gly238, and Arg276) while retaining penicillin and cefotaxime activity. Further, we have determined residues responsible for the extended-spectrum β-lactamase activity of the enzyme, defined as those necessary for high levels of cephalosporin, but not penicillin hydrolysis (Asn104, Asn106, Pro167, Lys234, Thr235, and Ser237). Finally, we show that omega loop residue (Pro167, Glu166, and Asn170), which are generally essential for ampicillin and cefotaxime activity, are not required for ceftazidime hydrolysis. In fact, substitutions at Pro167 increase ceftazidime resistance levels, indicating that the evolution of CTX-M resistance to ceftazidime is likely to occur through changes to the omega loop.

We have focused our experiments on a single penicillin and two cephalosporins. However, with the libraries in place, it will be of interest to expand the study to other β-lactams and β-lactamase inhibitors. We have shown that the determinants for catalysis can differ between β-lactam classes (penicillins versus cephalosporins), but also within a single class, as seen with cefotaxime and ceftazidime. Studies with other β-lactams can expand understanding of β-lactamase substrate specificity. Also, this approach can provide insights into the determinants of potency for inhibitors such as clavulanic acid, avibactam, and vaborbactam, which could identify resistance mutations and inform new inhibitor designs. Finally, this approach for examining substrate specificity can be applied to other enzyme classes which act on multiple different substrates, such as proteases, provided a genetic selection can be established to sort the mutants in the libraries based on function.

## Methods

**Bacterial strains and plasmids**. The *E. coli* strain XL1-Blue [*recA1, endA1, gyrA96, thi-1, hsdR17, supE44, relA1, lac*, [F9 *proAB lacIq lacZ*ΔM15, Tn10 (tetr)]] (Stratagene, Inc., La Jolla, CA) was used as the host for the construction of CTX-M-14 variants. The plasmid pTP123, which contains the gene for CTX-M-14 and chloramphenicol resistance[51], was used as the template for site-directed mutagenesis. The *E. coli* strain BL21(DE3) (*fhuA2 [lon] omp Tgal (λDE3) [dcm] ΔhsdS λ DE3 = λ sBamHIo ΔEcoRI-B int::(lacI::PlacUV5::T7gene1) i21 Δnin5*) was used as the host for protein expression of wild-type and mutant CTX-M-14 enzymes. A truncated, mature CTX-M-14 enzyme (CTX-M-14 residues 22-284) was cloned into a modified pET28a vector containing kanamycin resistance (Novagen), in which the thrombin recognition sequence was replaced with the tobacco etch virus (TEV) protease recognition sequence. Site-directed mutagenesis was performed on CTX-M-14-pET28a-TEV for the expression of single mutants for purification.

**Construction of CTX-M-14 single codon randomization libraries**. CTX-M-14 single codon randomization libraries were constructed by oligonucleotide-directed mutagenesis[52]. First, an *XhoI* restriction site was inserted near each of the 17 target codons by oligonucleotide-directed mutagenesis to allow the elimination of wild-type CTX-M-14 background. The *XhoI* restriction site was also designed to create a frameshift mutation, ensuring the DNA template is non-functional. The *XhoI* insert mutant was then used as the template for the randomization of each codon using partially overlapping (25-bp) primers for PCR amplification. The codon for the target residue was substituted for either NNS (where N is any of the four nucleotides, and S is G or C) or NNK (where K is G or T) such that the wild-type codon was excluded from the randomized library, further eliminating wild-type interference. The resulting mutagenesis reactions were treated with *DpnI* and *XhoI* restriction enzymes to eliminate non-mutagenized plasmids. The treated reactions were purified using Qiagen Minelute columns using Tris-Cl buffer, then transformed into *E. coli* XL1-Blue by electroporation. A minimum of 1000 colonies were pooled to constitute each library.

**Selection of library mutants against β-lactam antibiotics**. Approximately 50 ng of DNA from each library was transformed into *E. coli* XL1-Blue electrocompetent cells. The cells were recovered in pre-warmed LB for 1 h, then plated on LB agar containing 25 μg/mL chloramphenicol and allowed to grow at 37 °C overnight. The resulting colonies were pooled in cation-adjusted MHB (10 μg/mL Mg²⁺, 20 μg/mL Ca²⁺) containing 12.5 μg/mL chloramphenicol, and an aliquot of this naïve library was aside for DNA extraction (see next section for DNA extraction). The remaining culture was diluted to an OD₆₀₀ of 0.4 in

cation-adjusted MHB containing 12.5 μg/mL chloramphenicol. Then the cultures were diluted 1:100 into 1.2 mL round bottom plates, with either no β-lactam antibiotic, 32 μg/mL ampicillin, 0.25 μg/mL cefotaxime, or 0.50 μg/mL ceftazidime in cation-adjusted MHB containing 12.5 μg/mL chloramphenicol. Concentrations of antibiotics were determined to select for clones with near wild-type levels of β-lactamase activity. Cultures were allowed to grow for 20 h, shaking at 180 rpm at 37 °C.

**PCR amplification and NGS sequencing**. Plasmid DNA from overnight cultures after growth in the antibiotic of interest was isolated using a Plasmid Miniprep kit (Zymo Research) and used as template DNA for PCR amplification. A 160-bp region containing the targeted position was amplified, with a 6-bp ligation adapter and 8-bp barcode added to each end for downstream NGS sequencing. PCR products were purified from a 1.5% agarose gel using a QIAquick gel extraction kit (Qiagen), and the DNA concentration of each purified product was determined using a Nanoquant plate (TECAN). PCR products from each selected library and the naïve control were pooled into a single tube in equal concentrations, and Illumina deep sequencing was performed by Novogen Co., Ltd (read length 150 bp).

**Computational processing of NGS data**. Analysis of NGS deep sequencing FASTQ files was performed using a custom Python 3.0 script. DNA sequence reads were filtered for a minimum PHRED score of 20 for each nucleotide across the read using Trimmomatic[53]. The selection condition of each read was determined based on the 8-bp barcode, and the codon at the randomized position was identified by the 6-bp upstream and 6-bp downstream of the codon of interest. This yielded $6.9 \times 10^7$ total sequence reads, with an average of $8.1 \times 10^5$ per selection condition, per library (including the naïve control library that was not selected under β-lactam antibiotic). The number of times each amino acid occurred at the position of interest was counted to give totals used to calculate relative fitness (Supplementary Data 1). The total amino acid representation for each replicate was calculated separately. The sequence logo representing these results was created using the seqlogo function in MATLAB. For the sequence logo visualization, the amino acid distribution for each antibiotic selection condition was normalized to that of the naïve library to account for biases prior to antibiotic selection. Additionally, fitness values were calculated using Eq. 1, as shown and described in the Results and Discussion section.

**Protein expression and purification**. Wild-type CTX-M-14 and its variants were expressed using CTX-M-14-pET28a-TEV in *E. coli* BL21(DE3) cells for subsequent enzyme kinetics or X-ray crystallography experiments. *E. coli* containing CTX-M-14 expression plasmid were grown in an LB medium containing 25 μg/mL kanamycin. Expression of the His-tagged-CTX-M-14 protein was induced in mid-log-phase cultures with 0.2 mM IPTG at 23 °C for 20 h. The cells were pelleted and resuspended in lysis buffer (25 mM NaPO₄, 300 mM NaCl, 20 mM imidazole, pH 7.4). Sonication was used to disrupt cells, and cell debris was removed by centrifugation. Soluble fractions in the supernatant were loaded onto a column packed with Talon resin (Takara Bio), and His-tagged-CTX-M-14 was eluted with an imidazole gradient in the lysis buffer. Eluted protein was then concentrated using an Amicon® Ultra-15 Centrifugal Filter Unit (MilliporeSigma). Protein purity was visualized by SDS-PAGE followed by Coomassie Brilliant Blue (CBB) staining.

**Enzyme kinetics**. Michaelis–Menten steady-state kinetic parameters of CTX-M-14 β-lactamase variants for ampicillin, cefotaxime, and in some cases, ceftazidime were determined. Wavelength and extinction coefficients used for detection were: ampicillin, 235 nm, $\Delta\varepsilon = 900$ M⁻¹ cm⁻¹; cefotaxime, 264 nm, $\Delta\varepsilon = 7250$ M⁻¹ cm⁻¹; ceftazidime, 260 nm, $\Delta\varepsilon = 8660$ M⁻¹ cm⁻¹. Enzymatic reactions were performed at 25 °C in a buffer consisting of 50 mM sodium phosphate (pH 7.0) and 1 μg/mL BSA. Antibiotic hydrolysis was monitored with a DU800 spectrophotometer (Beckman Coulter). Initial hydrolysis velocities of substrate were plotted as a function of concentrations and fitted to the Michaelis–Menten equation by nonlinear regression using GraphPad Prism 5 (GraphPad Software) to obtain $K_M$ and $k_{cat}$ values, as shown in Fig. S5. Each $v/E_0$ value for a given substrate concentration was considered individually in this regression and in the error calculation described in the statistics and reproducibility subsection.

**Minimum inhibitory concentration (MIC) determination**. Wild-type CTX-M-14 and its variants were transformed via CTX-M-14-pTP123 vector into *E. coli* XL1-Blue electrocompetent cells. In addition, an empty pTP123 vector with no β-lactamase gene was transformed to be used as a negative control for antibiotic resistance. CTX-M-14 and variants were recovered in LB media for 1 h, then streaked on LB agar plates containing 12.5 μg/mL CMP. Following overnight incubation at 37 °C, one colony was picked for each variant and grown for 18 h in LB media containing 12.5 μg/mL CMP. For MIC determination, saturated *E. coli* cultures were diluted 12,000-fold into cation-adjusted (10 μg/mL Mg²⁺, 20 μg/mL Ca²⁺) Mueller Hinton Broth (MHB), then selected in 96-well plates containing increasing concentrations of ampicillin (8 to 80 μg/mL), cefotaxime (0.0625 to 64 μg/mL), or ceftazidime (0.0625 to 4 μg/mL). MIC plates were shaken for 18 h at

200 RPM, 37 °C. Following incubation, MICs were determined as the lowest antibiotic concentration that inhibited *E. coli* growth.

**X-ray crystallography**. The PEGs and PACT Suites from Molecular Dimensions were used to screen conditions for crystal growth using the hanging drop vapor diffusion method. A Mosquito robot (TTP Labtech Ltd., Melbourn, UK) was used to deposit nL volumes of liquid to set up crystal screens at protein concentrations of 25–50 mg/mL. The CTX-M-14 Y105W mutant enzyme (PDB: 7UON) was crystallized at pH 7.5 in 0.1 M HEPES, 25% w/v PEG 4000. Glycerol (15%) was used as a cryoprotectant. Diffraction data were collected at the Berkeley Center for Structural Biology at the Advanced Light Source synchrotron, beamline 8.2.1. The complete diffraction dataset was collected at 100 K from a single crystal at a wavelength of 1.00001 Å. All data were processed using iMosflm[54] and the CCP4i Suite[55]. Phaser[56] was used for molecular replacement using CTX-M-14 (PDB: 1YLT) as the model enzyme. Coot[57] was used to fit the model to the electron density and the phenix.refine[58] and REFMAC5[59] programs were used for refinement. Data collection and refinement statistics for Y105W are reported in Table 4. The Ramachandran favored for 7UON is 98.07%, while Ramachandran outliers were 0.38%. Figure panels containing crystal structures (Figs. 1, 4, 5, and S1) were created using the UCSF Chimera program[60]. The Polder map program in Phenix was used to create the omit map seen in Fig. 5b[61].

**Calculation of effective number of substitutions (*k\**)**. The effective number of substitutions (*k\**) for each of the 17 residue positions was calculated following selection under ampicillin, cefotaxime, or ceftazidime. The values were calculated according to Eq. 2:

$$S = - \sum_{i=1}^{k} p_i (\log_2 p_i)$$
$$k^* = 2^S \tag{2}$$

In the equation, *S* is entropy, $p_i$ is the frequency with which a given amino acid (*i*) appears at a position, and *k* is the number of different amino acid types that appear at a position. A $k^* = 1$ indicates that, functionally, only one amino acid is observed at this position, while $k^* = 20$ indicates that all amino acids occur at equal frequencies[34,35].

**Statistics and reproducibility**. Next-generation sequencing reads were filtered based on the PHRED quality score, as described in the NGS processing subsection. Selection and sequencing for each library was performed one to three times. DNA libraries were transformed into *E. coli*, pooled, and the resulting pooled bacteria culture was selected in β-lactam-containing media, constituting biological replicates. The biological replicate values are displayed as red dots in Fig. 6 and S2–4, and the mean of those values is displayed as a black line. In cases where the frequency of the mutant was equal to 0 following selection, the replicate was excluded from fitness measurements. In cases where all replicates were excluded, the fitness was not quantifiable and therefore not included, though we can conclude the fitness is very low.

MIC values in Table 2 represent at least two replicates for every CTX-M-14 variant against each antibiotic ($n \geq 2$). Figure S5 shows the fits of data to the Michaelis–Menten equation and plotted points represent $v/E_0$ values (initial velocity divided by enzyme concentration) for a given antibiotic concentration and includes all data points, including replicates and singly determined values. The standard error of steady-state kinetic parameters, $K_M$ and $k_{cat}$, as shown in Tables 2, 3, were calculated in GraphPad Prism based on the standard error of least squares nonlinear regression of the curve fit, which represents the average distance of the data values from the nonlinear regression line. The initial rate values can be found in Supplementary Data 2. The $K_M$ and $k_{cat}$ standard error values were used to calculate the catalytic efficiency ($k_{cat}/K_M$), and the error was propagated according to Eq. 3:

$$SE(k_{cat}/K_M) = \sqrt{(SE k_{cat}/k_{cat})^2 + (SE K_M/K_M)^2} \tag{3}$$

**Reporting summary**. Further information on research design is available in the Nature Portfolio Reporting Summary linked to this article.

## Data availability
The CTX-M-14 Y105W structure information has been deposited in the Protein Data Bank and assigned as PDB ID 7UON. Sequencing counts following PHRED filtering used to generate fitness and frequency values are available in Supplementary Data 1. Calculated residue fitness, frequency, *k\** values, and steady-state kinetics datasets represented in this article are available in Supplementary Data 2. Plasmid expression constructs of β-lactamases are available upon request.

## Code availability
The custom Python 3.0 script used to process NGS FASTQ files and determine fitness and frequency values is available upon request.

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

## Acknowledgements

This work was funded by NIH grant AI32956 to T.P. and Welch Foundation grant Q1279 to B.V.V.P. A.J. was funded by NIH training grant T32 GM120011. The ALS-ENABLE beamlines are supported in part by the National Institutes of Health, National Institute of General Medical Sciences, grant P30 GM124169-01. The Advanced Light Source is a Department of Energy Office of Science User Facility under Contract No. DE-AC02-05CH11231.

## Author contributions

A.J. performed experiments, analyzed data, and wrote and edited the manuscript. L.H. analyzed crystallography data, performed data visualization, and edited the manuscript. B.S. performed X-ray diffraction experiments, analyzed crystallography data, and edited the manuscript. J.V.R. performed biochemical experiments and edited the manuscript. B.V.V.P. supervised structure determination, obtained funding, and edited the manuscript. T.P. supervised experiments, analyzed data, obtained funding, and edited the manuscript.

## Competing interests

The authors declare no competing interests.
