## [Peer Review File · Communications Biology]

Reviewers' comments:

Reviewer #1 (Remarks to the Author):

The manuscript entitled, "Mapping the Determinants of Catalysis and Substrate Specificity of the Antibiotic Resistance Enzyme CTX-M β -lactamase" by Judge et al. describes a detailed and careful study using codon randomization, antibiotic selection, and deep sequencing to probe which active site residues in CTM-M are required for hydrolysis of cefotaxime, ampicillin, and ceftazidime. The experiments in this manuscript were very well done and proper controls were done. This work is very important because the results could be used to guide future inhibitor design efforts for CTX enzymes. There are a few minor corrections in the manuscript, but I strongly suggest that the authors expand the conclusions a little bit to include how these results could be used to guide future studies. I also suggest that the authors explain how these studies could be used in other enzyme systems to understand substrate binding and specificity and possibly drug design.

Minor issues:

1. Line 40, Change " β -lactamase enzymes" to " β -Lactamases.."
2. Line 81, I do not believe that references 15 and 22 should be superscripted

Reviewer #2 (Remarks to the Author):

This is an interesting study on catalytic mechanisms of CTX-M β -lactamases. The manuscript is well written and includes a set of detailed experiments. Mutagenesis and deep sequencing has been performed using recent technologies (e.g. NGS). The role of single amino acids in the hydrolysis of β -lactams has been evaluated by kinetic parameters determination and structural experiments.

This study is worthy of publication.

Reviewer #3 (Remarks to the Author):

This manuscript examines the residues of CTX-M-14 enzyme which are involved in the catalysis of ampicillin and two cephalosporins, cefotaxime and ceftazidime. The authors used codon randomization to compare the fitness of different variants of important residues without antibiotic selection pressure and in presence of ampicillin, cefotaxime and ceftazidime. Some of the variants displaying better fitness were analyzed for their ability to hydrolyze these antibiotics in cells and in vitro. Additionally, comparison of the occurrence of certain variants in presence of antibiotics allowed the authors to make general conclusions about the roles of residues in catalysis. An important conclusion was made about the active site omega-loop modification required for the evolution of ceftazidime hydrolysis.

The work appears to have been performed thoroughly and the conclusions are mostly supported by the data (see below for some issues). It is original, but the broader context of work on other β -lactamases in which many the same mutants were studied should be given (see below for examples). The paper is primarily of interest to other researchers in the field of β -lactamases, and thus, antibiotic resistance. Publication is recommended after addressing the issues below.

Issues to be addressed.

Reference to relevant literature is not always given. In relation to the Glu166 mutations (line 175),

there is much literature to be found for similar class A lactamases, e.g. DOI 10.1038/s41598-020-66431-w or Journal of Biological Chemistry, 271, 10482-10489. Similarly, for the mutants of position 105 (line 200), DOI: 10.1074/jbc.M407606200. There's more that is relevant to quote.

Reference formatting is different throughout the introduction, which leads to some references being incorrect (for example, line 83 contains references 21 and 28, while both articles do not have any information about the topic discussed in the text; or references 18-19 mentioned before reference 16; line 434 has "(source)" written in the text without the source itself, etc.).'

The discussion of the key catalytic residues (lines 152-166) indicates that there were 6 residues identified as non-variable with any substitutions leading to a significant reduction in fitness. But according to the supplementary data substitution of Asn132 to Ser can occur without negative effect on ampicillin fitness, N132 is even labeled as flexible in figure 4, panel A. The conclusions made by authors are still in line with the presented data, but this one "good" variant should be acknowledged.

In line 221/222, it is stated that "...shorter side chain (Ser106) eliminates the hydrogen bond interaction between Val103 and residue 106". However, in ref. 16 (from the same group) it is stated that a H-bond with the carbonyl of Val103 is still present and PDB entry 6CYK shows indeed a short distance of 2.5 Å between the O(g) and CO. Thus the statement is not correct, only the H-bond with the NH is lost due to the mutation to Ser.

The discussion of the Arg276 sequencing results (lines 239-243) states that "it is clear that many different amino acids are consistent with high levels of function for both ampicillin and cefotaxime", however, figures S1 and S2 as well as the supplementary table, show that there is a clear preference for only a few type of residues (mostly very conservative substitutions, which is an important result), while the rest display relative fitness of less than -1/-2. Furthermore, lines 242-243 reference studies that showed that Arg276 substitutions have little effect on CTX-M activity, while reference 24 shows a significant decrease of MIC values for some antibiotics for many mutants.

The kinetic parameters reported in the main text table 2 are different from those reported in the supplementary (Figure S4).

Deep sequencing of the Asp240 residue resulted in two variants D240V and D240P being studied for their kinetics and MIC values. The authors say that these variants were chosen because they were the most enriched variants after selection with ampicillin and cefotaxime (lines 228-230), respectively, while the most enriched variant for both selections was actually D240G, which was not studied within this research line probably because it was already studied before by the same group. In that light the choice of D240V and D240P over D240G is a good choice, but D240G should be mentioned as the most occurring variant.

Lines 249-251 identify residues for which substitutions were found that retain high ampicillin fitness but decrease fitness for cefotaxime. One of these variants is Gly238Cys, however, figures S1-S2 show that this substitution result in only a slightly decreased fitness with a big error bar for cefotaxime, which is a result of one screening out of three being significantly lower than other two. It also does not go from flexible to stringent according to k^* values (panel A and B of figure 4). So the conclusions about it's importance for cefotaxime but not ampicillin are not supported by the data. Another variant Thr235Ser shows exactly the same effect on fitness (slight decrease with large error bar coming from one screening having much lower count than other two) for ampicillin and is discussed as retaining the activity. So these two variants should be treated similarly in terms of conclusions.

The message that the omega-loop evolution is required for the improvement of ceftazidime hydrolysis is interesting, especially combined with the shown decrease in hydrolysis of another antibiotics. The article may benefit in strengthening this point by referencing more literature about variants found in clinical isolates with ceftazidime/ceftazidime-avibactam resistance, where omega-loop mutations are

often observed. As it is always a good sign when in-lab evolution corresponds well with the natural evolution. (e.g., Compain, Dorchene and Arthur, AAC, 2018; Hobson et al., AAC, 2022).

Some of the positions for ampicillin and cefotaxime were screened only once, how reliable are the results in that case? Also, as the positions in the naïve library were all screened in triplicate, how the analysis of the fitness was performed in the case of the positions done once, was the fitness calculated with one of the naïve library repeat or with the average of three?

In Figure 3 the stretched letters are hard to read, that should be made clearer.

In the caption of Figure 5, last line, "Tyr105" should be "Trp105".

Reviewer #4 (Remarks to the Author):

1. Brief Summary:

The paper Mapping the determinants of catalysis and substrate specificity of the antibiotic resistance enzyme CTX-M beta-lactamase by Judge et al shows an interesting study about the specificity, gain, and loss of function of CTX-M-14. Using CTX-M-14 as a model, the authors performed randomized codon libraries, deep sequencing, enzymatic activity, and structural studies to determine which residues are essential for the specificity of this class of beta-lactamases.

2. Overall Impression of the work

Beta lactamases are one of the main mechanisms of antimicrobial resistance and they are encoded mostly on mobile elements, which facilitates their spread. The use of wide-spectrum antibiotics causes the evolution of these enzymes and most of them can hydrolyze ceftazidime, the last resort of antibiotics to treat infections caused by gram-negative bacteria. The study of beta-lactamases and its evolution is crucial for developing new drugs to treat infections caused by bacteria and cope with antimicrobial resistance.

The work presented here has a high value because it described that the changes in the omega-loop, a structural element that is indispensable for catalysis, are often required for ceftazidime hydrolysis. However, this could be detrimental to the activity of the enzyme to hydrolyze other substrates such as penicillin. The use of deep sequencing and randomized could provide an interesting tool to determine whether infections caused by bacteria that are resistant to ceftazidime, might be treated with penicillin.

3. Specific comments with recommendations for addressing each comment

Based on the value of the study as well as the approach, I would like to recommend the acceptance of the paper entitled Mapping the determinants of catalysis and substrate specificity of the antibiotic resistance enzyme CTX-M beta-lactamase with minor corrections:

- Please format references according to the specifications of communications biology.
- The abstract does not have a clear hypothesis for the study. I would like to suggest adding a small hypothesis.
- Add references to line 48.
- One of the weaknesses of the discussion is the relevance of these mutations and their prevalence in nature or clinical isolates. I think that the discussion could benefit from these comparisons.
- Fig.1. Shows the structure of CTX-M-14 and the 17 active site residues. It could be beneficial for the readers also highlight the residues that form the omega-loop and the catalytic serine70.
- Fig. 5. Please add labels to the residues and panels A and B.

Response to Reviewers' Comments:

Reviewer #1 (Remarks to the Author):

	Comment	Response
1	There are a few minor corrections in the manuscript, but I strongly suggest that the authors expand the conclusions a little bit to include how these results could be used to guide future studies. I also suggest that the authors explain how these studies could be used in other enzyme systems to understand substrate binding and specificity and possibly drug design.	We have revised the conclusions section to include the possible future uses of this approach to expand understanding of b-lactamase substrate specificity to other b-lactams and to use the approach to examine pathways to b-lactamase inhibitor resistance and informing inhibitor design. In addition, we have discussed how the approach could be generalized to understanding substrate specificity for other enzyme classes, such as proteases.
2	Line 40, Change “b-lactamase enzymes” to “b-Lactamases..”	The suggested language change on line 40 was made.
3	Line 81, I do not believe that references 15 and 22 should be superscripted	The Nature referencing style has been implemented throughout the document, which includes superscripted reference numbers rather than parentheses.

Reviewer #2 (Remarks to the Author):

	Comment	Response
1	This is an interesting study on catalytic mechanisms of CTX-M beta-lactamases. The manuscript is well written and includes a set of detailed experiments. Mutagenesis and deep sequencing has been performed using recent technologies (e.g. NGS). The role of single amino acids in the hydrolysis of beta-lactams has been evaluated by kinetic parameters determination and structural experiments. This study is worthy of publication.	We thank the reviewer for the positive comments on the manuscript.

Reviewer #3 (Remarks to the Author):

Issues to be addressed.

	Comment	Response
1	Reference to relevant literature is not always given. In relation to the Glu166 mutations (line 175), there is much literature to be found for similar class A lactamases, e.g. DOI 10.1038/s41598-020-66431-w or Journal of Biological Chemistry, 271, 10482-10489. Similarly, for the mutants of position 105 (line 200), DOI: 10.1074/jbc.M407606200. There's more that is relevant to quote.	The requested additional citations on the E166Y mutation in TEM-1 were added (now contained in lines 180 and 181). The suggested citation on position 105 was added (now contained in line 220). As the reviewer notes, there was more relevant to cite, which are highlighted throughout the text, including literature on: E166Y in PenP (line 181), N132S in Streptomyces albus G β -lactamase (line 184), Arg276 mutations in CTX-M-1 (line 267), and several mutations from clinical isolates in the ceftazidime omega loop section (lines 330-381).

	Comment	Response
2	Reference formatting is different throughout the introduction, which leads to some references being incorrect (for example, line 83 contains references 21 and 28, while both articles do not have any information about the topic discussed in the text; or references 18-19 mentioned before reference 16; line 434 has "(source)" written in the text without the source itself, etc.).'	As pointed out, several references were not correctly logged in the citation manager and were therefore mis-numbered. These have been carefully checked and should accurately reflect the source material. Additionally, the error on what was line 434 (now line 512) is corrected and contains the appropriate reference to Stiffler, et. al. 2015.
3	The discussion of the key catalytic residues (lines 152-166) indicates that there were 6 residues identified as non-variable with any substitutions leading to a significant reduction in fitness. But according to the supplementary data substitution of Asn132 to Ser can occur without negative effect on ampicillin fitness, N132 is even labeled as flexible in figure 4, panel A. The conclusions made by authors are still in line with the presented data, but this one "good" variant should be acknowledged.	The reviewer is correct to point out this notable exception, which is now included in the discussion. This paragraph is now on lines 182-194, for your reference.
4	In line 221/222, it is stated that "...shorter side chain (Ser106) eliminates the hydrogen bond interaction between Val103 and residue 106". However, in ref. 16 (from the same group) it is stated that a H-bond with the carbonyl of Val103 is still present and PDB entry 6CYK shows indeed a short distance of 2.5 Å between the O(g) and CO. Thus the statement is not correct, only the H-bond with the NH is lost due to the mutation to Ser.	This statement was erroneous and has been removed. The discussion comparing the N106S and N106Q mutations (lines 223-245) is also updated to include a more nuanced description, including information from a recently published paper from our group.
5	The discussion of the Arg276 sequencing results (lines 239-243) states that "it is clear that many different amino acids are consistent with high levels of function for both ampicillin and cefotaxime", however, figures S1 and S2 as well as the supplementary table, show that there is a clear preference for only a few type of residues (mostly very conservative substitutions, which is an important result), while the rest display relative fitness of less than -1/-2. Furthermore, lines 242-243 reference studies that showed that Arg276 substitutions have little effect on CTX-M activity, while reference 24 shows a significant decrease of MIC values for some antibiotics for many mutants.	The discussion on position 276 was updated to include a more thorough analysis which, as the reviewer points out, includes the notable finding that there is a clear preference in amino acid type at the position (now lines 263-273). Additionally, Arg276 is now described as allowing "a conservative range of mutations" rather than being freely substitutable in lines 414-415.
6	The kinetic parameters reported in the main text table 2 are different from those reported in the supplementary (Figure S4).	a. Typos in the supplementary Figure S5 (formerly Fig. S4) were corrected to match the experimental values reported in Table 2. These include the k_{cat} of cefotaxime for the wild-type enzyme (which was changed from 75 to 76) and the k_{cat} of cefotaxime for the N106Q variant (which was changed from 37 to 27). The updated figure can be seen at the bottom of this document. b. In addition, the Michaelis-Menten curves for the S130T variant were previously omitted and are now included in Fig. S5.

		The data was originally plotted and evaluated by a previous lab member, but that file was not recovered; for this reason, the raw values were re-plotted and re-interpreted. The overall interpretation for S130T (that it has partial function against cefotaxime, but very little function for ampicillin or ceftazidime hydrolysis) is unchanged, though the kinetic parameters are somewhat different. c. Finally, references were added to Table 3 for previously published kinetic parameters, along with the explanation of error estimates, which were originally only included below Table 2.
7	Deep sequencing of the Asp240 residue resulted in two variants D240V and D240P being studied for their kinetics and MIC values. The authors say that these variants were chosen because they were the most enriched variants after selection with ampicillin and cefotaxime (lines 228-230), respectively, while the most enriched variant for both selections was actually D240G, which was not studied within this research line probably because it was already studied before by the same group. In that light the choice of D240V and D240P over D240G is a good choice, but D240G should be mentioned as the most occurring variant.	The reviewer's conclusion is correct, and the reasoning for selecting D240P and D240V for further study has been clarified to say: "A previously studied CTX-M variant, D240G, was found to be the most enriched residue at the position following selection in both ampicillin and cefotaxime²⁰. Among the remaining substitutions tested here, most of which produced a functional enzyme based on sequencing results (Figs. S2 and S3), D240P and D240V were predicted to have the highest activity and were chosen for further validation." (now found in the paragraph on lines 246-262).
8	Lines 249-251 identify residues for which substitutions were found that retain high ampicillin fitness but decrease fitness for cefotaxime. One of these variants is Gly238Cys, however, figures S1-S2 show that this substitution result in only a slightly decreased fitness with a big error bar for cefotaxime, which is a result of one screening out of three being significantly lower than other two. It also does not go from flexible to stringent according to k^* values (panel A and B of figure 4). So the conclusions about its importance for cefotaxime but not ampicillin are not supported by the data. Another variant Thr235Ser shows exactly the same effect on fitness (slight decrease with large error bar coming from one screening having much lower count than other two) for ampicillin and is discussed as retaining the activity. So these two variants should be treated similarly in terms of conclusions.	The reviewer is correct that the fitness data does not support the claim that position 238 has differential sequence requirements for ampicillin and cefotaxime. For this reason, we have updated our conclusions, classifying G238C as a partial-function mutant and position Gly238 as a flexible position that allows for this mutation while retaining some ampicillin and cefotaxime hydrolysis function. Gly238 is now discussed on lines 274-286.
9	The message that the omega-loop evolution is required for the improvement of ceftazidime hydrolysis is interesting, especially combined with the shown decrease in hydrolysis of another antibiotics. The article may benefit in strengthening this point by referencing more literature about	The suggested sources, among others, have been added to the subsection discussing the role of the omega-loop in ceftazidime hydrolysis (now lines 330-381). We expanded the discussion on tradeoffs between ceftazidime and ceftazidime-

	variants found in clinical isolates with ceftazidime/ceftazidime-avibactam resistance, where omega-loop mutations are often observed. As it is always a good sign when in-lab evolution corresponds well with the natural evolution. (e.g., Compain, Dorchene and Arthur, AAC, 2018; Hobson et al., AAC, 2022).	avibactam activity vs. cefotaxime/penicillin activity to include other beta-lactamases. As suggested by the reviewer, this strengthens our conclusions on the subject.
10	Some of the positions for ampicillin and cefotaxime were screened only once, how reliable are the results in that case? Also, as the positions in the naïve library were all screened in triplicate, how the analysis of the fitness was performed in the case of the positions done once, was the fitness calculated with one of the naïve library repeat or with the average of three?	a. Further validation of a subset of mutations within these libraries support our conclusions. An example of such a library is N106X selection against ampicillin, which is further validated in the N106Q mutant MICs (Table 1) and steady-state kinetics (Table 2). Similarly, P167E was validated to sufficiently support our conclusions. b. The representation, or fraction, of each residue within the naïve libraries was calculated individually for each replicate. This was done to account for any biases in library composition after DNA was transformed into bacteria, which was the step immediately before pooling of E. coli colonies and selection in β -lactam antibiotic. The handful of selection conditions that only have one replicate had their fitness calculated in the same way—from the number of colonies, and thus total mutants, from the naïve library immediately prior to selection. This has been clarified in the methods section in the paragraph from lines 468-480.
11	In Figure 3 the stretched letters are hard to read, that should be made clearer.	We have re-formatted the figure to have multiple rows so that the letters could be made wider and less stretched. The revised figure can be seen below in the updated figures section.
12	In the caption of Figure 5, last line, “Tyr105” should be “Trp105”.	The correction to Figure 5 was made and can be seen below.

Reviewer #4 (Remarks to the Author):

Specific comments with recommendations for addressing each comment

	Comment	Response
1	Please format references according to the specifications of communications biology.	The references were re-formatted to fit the standard Nature referencing style, as requested.
2	The abstract does not have a clear hypothesis for the study. I would like to suggest adding a small hypothesis.	The abstract has been revised to include a hypothesis and overall interpretation in reference to the hypothesis (lines 22-35).

3	Add references to line 48.	A reference was added to line 48 to support the statement that “The widespread use of these antibiotics, however, has led to the emergence of extended-spectrum β -lactamases (ESBLs), which are defined by their ability to hydrolyze oxyimino-cephalosporins”.
4	One of the weaknesses of the discussion is the relevance of these mutations and their prevalence in nature or clinical isolates. I think that the discussion could benefit from these comparisons.	The discussion on ceftazidime resistant mutations has been expanded, including omega loop mutations found in ceftazidime and ceftazidime-avibactam resistant clinical isolates (now contained in lines 330-406).
5	Fig.1. Shows the structure of CTX-M-14 and the 17 active site residues. It could be beneficial for the readers also highlight the residues that form the omega-loop and the catalytic serine70.	Figure S1 was added to highlight key active site structures for readers to reference. The figure can be seen below in the updated figures section.
6	Fig. 5. Please add labels to the residues and panels A and B.	Labels are now included for the residues in panels A and B, which can be seen below.

Updated figures:

Figure S5 (formerly Fig. S4) was updated in response to Reviewer #3, comment 6. Typos were corrected to match the experimental values reported in Table 2. These include the k_{cat} of cefotaxime for the wild-type enzyme (which was changed from 75 to 76) and the k_{cat} of cefotaxime for the N106Q variant (which was changed from 37 to 27). In addition, the Michaelis-Menten curves for the S130T variant were previously omitted and are now included in Fig. S5.

Figure 3 was updated in response to Reviewer #3, comment 11. The figure was re-formatted to include multiple rows so that the letters could be made wider and less stretched.

Figure 5 was updated in response to Reviewer #3, comment 12. In the last line of the caption, “Tyr105” was changed to “Trp105.”

Figure S1 was created in response to Reviewer #4, comment 5. The figure highlights key active site structures for readers to reference.

Figure S5. Each curve was fitted to the Michaelis-Menten model in Graphpad Prism to determine steady-state kinetic parameters (k_{cat} , K_M , and k_{cat}/K_M) described in Tables 2 and 3. Mutant CTX-M-14 enzymes were tested against cefotaxime (CTX), ampicillin (AMP), or ceftazidime (CAZ).

Figure 3. Sequence logos representing the relative frequency of each residue at each of the 17 randomized active site positions. The height of each amino acid (single letter code) is proportional to its frequency following selection in ampicillin (amp), cefotaxime (ctx), or ceftazidime (caz).

Figure 5. CTX-M-14 Y105W enzyme structure. (A) CTX-M-14 Y105W enzyme structure (blue, PDB: 7UON) determined at 1.35 Å superimposed on wild-type CTX-M-14 (tan, PDB: 1YLT). Note that the tryptophan assumes an essentially identical position as the wild-type tyrosine. (B) Polder map of 7UON (3.0σ), constructed with Trp105 omitted, which demonstrates the certainty of the positioning of Trp105.

Figure S1. Key structural motifs of CTX-M-14 β -lactamase (PDB: 1YLT), including **(a)** the Ω -loop containing residues 166-170, **(b)** the 103-106 loop, and **(c)** the β 3 strand containing residues 234-240.

REVIEWERS' COMMENTS:

Reviewer #3 (Remarks to the Author):

The concerns raised have been addressed thoroughly and adequately. The manuscript is acceptable as is.